# Evaluation of the Amino Acid Composition and Content of Organic Acids of Complex Postbiotic Substances Obtained on the Basis of Metabolites of Probiotic Bacteria *Lacticaseibacillus paracasei* ABK and *Lactobacillus helveticus* H9

**Irina Vladimirovna Rozhkova, Elena Anatolyevna Yurova \* and Victoria Alexandrovna Leonova**

Federal State Autonomous Scientific Institution "All-Russian Dairy Research Institute" (FGANU "VNIMI"), 115093 Moscow, Russia; i_rozhkova@vnimi.org (I.V.R.); v_leonova@vnimi.org (V.A.L.)
* Correspondence: elena.a.yurova@yandex.ru

**Abstract:** In this article, the probiotic strains of *Lacticaseibacillus paracasei* ABK and *Lactobacillus helveticus* H9 were cultured in reconstituted skim milk (RSM medium) and MRS broth, and the cell biomass was removed at the end of fermentation in order to obtain postbiotic substances. In postbiotics, the composition of total amino acids, and also the content of free amino acids and organic acids were analyzed. It was shown that in all RSM-based postbiotic substances the concentration of all free amino acids increased. On the contrary, in the MRS-based postbiotics free amino acids were mostly consumed during fermentation; however, a substantial, two-fold, decrease in methionine concentration was observed in postbiotics obtained with *L. paracasei* ABK. Both *L. paracasei* ABK and *L. helveticus* H9 strains showed change in their fermentation profile from homofermentative in MRS broth to mix-acid fermentation in RSM medium. Both strains produced lactic acid in the investigated media and produced lactate together with acetate in RSM. *L. helveticus* H9 additionally synthesizes succinic acid on both media. Thus, it has been shown that RSM is more preferable than MRS for fermentation with *L. paracasei* ABK and *L. helveticus* H9 for obtaining postbiotics enriched with free amino acids and organic acids.

**Keywords:** *Lacticaseibacillus paracasei*; *Lactobacillus helveticus*; probiotics; postbiotics; amino acids; organic acids; capillary electrophoresis methods

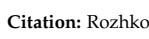



## 1. Introduction

One of the most important tasks for improving the nutrition of the population is to increase the output of mass consumption products with high nutritional and biological value. At the same time, nutrition should not only satisfy the physiological needs of a person, but also perform preventive and therapeutic functions [1]. Products with functional properties include fermented milk products enriched with probiotic microorganisms. Since symbiotic microorganisms of the gastrointestinal tract (GIT) and their metabolites are involved in many functions of the host organism, such products can be used to correct functional disorders of various organs and systems of the body. However, in some cases, therapy using living cells of probiotics is undesirable, in particular, in pancreatitis [2]. In addition, the products of their metabolism and structural components, known as postbiotics, can become an addition to probiotic microorganisms. These components are able to have a positive effect on homeostasis and metabolic signaling pathways of the body that affect physiological, immunological and neurohormonal metabolic responses.

Today, the main directions in the development of means for correcting disorders of the microbiota of the human gastrointestinal tract are the improvement of traditional probiotics and the production of postbiotics based on them [2]. At the same time, post-biotics can potentially be used in conjunction with probiotic microorganisms. In addition, postbiotics

are characterized by high digestibility and resistance to the GIT conditions. Due to this, their potential use for the development of products with functional properties is possible.

Postbiotics reach the colon by 95–97% unchanged. Due to the presence of biologically active compounds, postbiotics enter into metabolic reactions immediately upon entering the gastrointestinal tract. This eliminates the problem of colonization of the gastrointestinal tract, which is characteristic of probiotics and caused by antagonism between probiotic cultures and resident representatives of the microbiota [3].

Postbiotics optimize the functioning of the gastrointestinal tract by influencing commensal microorganisms. In addition, according to data from the literature, a number of postbiotics have immunomodulatory, antitumor and hypocholesterolemic effects [4]. According to the authors of [5], the benefits of using postbiotics may lie in a direct effect on host cells and act indirectly, contributing to an increase in beneficial microbial strains and inhibition of the development of negative ones. There is an opinion that the most important beneficial effect of postbiotics is their anti-inflammatory and antioxidant properties, in particular, due to the content of bioactive peptides and organic acids [6].

The main metabolites of probiotic lactic acid bacteria (LAB) include organic acids, short-chain fatty acids, exopolysaccharides, vitamins, amino acids, enzymes, bacterioins, etc. [7].

The presence of lactic acid in the human GIT increases the phagocytic activity of leukocytes, and together with acetic acid is also the basis of the antimicrobial activity of lactic acid bacteria [8,9]. Lactic acid and lactate can be converted to butyrate, which is the main source of carbon for the intestinal microbiota [10]. Acetate enters the blood vessels, being absorbed in the gastrointestinal tract, and with the bloodstream enters the peripheral tissues, where it participates in metabolic processes [11]. According to the results of studies on laboratory animals, a high content of acetate in the diet increases the body's resistance to infections [12]. Propionate exhibits an anti-inflammatory effect in vivo, and has a statin-like effect by inhibiting the cholesterol synthesis pathway [12].

Amino acids, being monomers of protein structures necessary for the body to work, take part in the synthesis of nitrogenous bases of nucleic acids, a number of hormones and other biologically active compounds. The provision of the body with amino acids determines its normal functioning, performance and resistance to adverse environmental factors [13–15].

Amino acids are classified into essential and non-essential. Non-essential amino acids are synthesized in the body from other amino acids during metabolism, while essential ones cannot be synthesized due to the lack of genes encoding their biosynthesis. These amino acids are formed by the symbiotic microbiota or are ingested with food. The following amino acids are considered essential for humans: phenylalanine, valine, threonine, tryptophan, isoleucine, methionine, leucine and lysine; for children, cysteine, tyrosine, histidine and arginine are additionally required [14,16] due to which it enters into a number of metabolic reactions providing the synthesis of choline and phospholipids [14].

Tryptophan is a precursor of the growth stimulator serotonin, which performs a neurotransmitter function and is responsible for the health of the nervous system and emotional behavior of a person [17,18]. It and its metabolites are believed to be one of the main elements of the immune balance in the intestine [19,20]. Glycine is a component of collagen, and in its free form it functions as an inhibitory neurotransmitter in some parts of the brain and spinal cord [21].

Products do not always contain enough amino acids to meet human needs. Studies of food raw materials have shown that it is necessary to additionally use natural essential and non-essential amino acids in human nutrition in order to meet the daily human needs for its normal functioning [15,22].

When obtaining amino acids for food and medical purposes, technologies based on microbial synthesis are used, which is due to the economic advantages and environ-mental friendliness of these methods in comparison with chemical synthesis. Genetically modified strains of *Corynebacterium glutamicum* and *Escherichia coli* are usually used as producers.

However, the use of these species of microorganisms causes controversy regarding their safety, which prompts the search for new amino acid producers [23].

In the studies of Toe, C.J., et al. [24] demonstrated the LAB's ability to form extracellular amino acids, including those that are indispensable (essential) in human nutrition. Due to this, postbiotics derived from LAB can potentially be used as sources of amino acids [23,24].

Therefore, the additional enrichment of fermented milk products with probiotic bacteria metabolism products and, in particular, with amino acids and organic acids, is of great interest to scientists from different countries and is a relevant and sought-after direction.

## 2. Materials and Methods

### 2.1. Preparation of Postbiotic Substances

Cultures of *L. helveticus* H9 and *L. paracasei* ABK were obtained from the Collection of the Central Laboratory of Microbiology of FGANU VNIMI. *L. helveticus* H9 was isolated from the gastrointestinal tract of a healthy person; *L. paracasei* ABK was isolated from a natural association of microorganisms—kefir grain. Bacterial strains were maintained in a de Man, Rogosa and Sharpe (MRS) broth (Merck, Germany) supplemented with 20% glycerol at −80 °C until use. Before the experiments, all strains were subcultured twice for 16 h at the optimum temperature for each strain (30 °C for *L. paracasei* ABK; 37 °C for *L. helveticus* H9) on MRS-broth and sterilized reconstituted skim milk (RSM) (12.5% *w/v*).

To obtain postbiotic substances the *L. helveticus* H9 and *L. paracasei* ABK strains were added in the amount of 3% (*vol/vol*) and cultivated on the nutrient medium MRS broth (MRS_Lh and MRS_Lp) and nutrient medium RSM (RSM_Lh and RSM_Lp) for 24 h at the optimum temperature for each strain (30 °C for *L. paracasei* ABK; 37 °C for *L. helveticus* H9) without aeration and without mixing.

At the end of incubation on MRS broth, the number of viable cells was $2.0 \times 10^8$ and $4.0 \times 10^7$ CFU mL$^{-1}$ at pH 3.88 and 4.98 for *L. helveticus* H9 and *L. paracasei* ABK, respectively. At the same time, at the end of fermentation on RSM, the number of viable cells was $1.2 \times 10^8$ and $3.3 \times 10^8$ CFU mL$^{-1}$ at pH 3.95 and 4.50 for *L. helveticus* H9 and *L. paracasei* ABK, respectively.

The cell biomass (in case MRS broth) and cell biomass with coagulated caseins (in case RSM) was separated by centrifugation at 6000 rpm for 15 min at 4 °C using a Hettich Rotanta 46 R centrifuge (Beverly, MA, USA). The supernatants (postbiotic substances) were filtered through MF-Millipore® Membrane Filter, 0.22 µm pore size (Sigma-Aldrich, St. Louis, MO, USA).

The obtained postbiotic substances (cell-free supernatant MRS broth fermented with *L. helveticus* H9 and *L. paracasei* ABK strains—MRS_Lh and MRS_Lp, respectively; cell-free supernatant RSM fermented with *L. helveticus* H9 and *L. paracasei* ABK strains—RSM_Lh and RSM_Lp, respectively) were examined for the total amino acid composition, free amino and organic acids content.

### 2.2. Determination of Amino Acid Composition, Free Amino and Organic Acids Content

The analysis of total amino acids composition and content of free amino acids and organic acids in control media (RSM and MRS) and postbiotic substances (RSM_Lh and RSM_Lp; MRS_Lh and MRS_Lp) were carried out with a capillary electrophoresis system "Kapel-105M" (Lumex Ltd., St. Petersburg, Russia) equipped with a spectrophotometric detector and with a special geometry quartz capillary (pore inner diameter of 50 µm and a total length of 75 cm for amino acids and pore inner diameter of 75 µm and a total length of 60 cm for organic acids). Electropherograms were processed using specialized Elforun® 205 software (S-Pb, St. Petersburg, Russia).

When determining the amino acid composition, the samples were preliminarily subjected to acid and alkaline (for tryptophan) hydrolysis in order to transfer amino acids from protein-bound forms to free ones. For all amino acids except tryptophan, phenylisothiocarbamyl derivatives were obtained, which were separated and quantified by capillary electrophoresis. For tryptophan, a direct determination was applied without obtaining a

TFA derivative. Tryptophan was determined using a borate buffer solution, voltage +25 kV, ultraviolet detection at 219 nm. Glutamic acid, aspartic acid and cystine were determined in phosphate buffer solution with the addition of β-cyclodextrin, at a voltage of +25 kV, a pressure of 50 mbar and UV detection at 254 nm. To determine the remaining amino acids (arginine, lysine, tyrosine, phenylalanine, histidine, leucine + isoleucine, methionine, valine, hydroxyproline, proline, threonine, serine, alanine, glycine), a method similar to the previous one was used, but without applying pressure.

When determining organic acids, the samples were preliminarily diluted with distilled water, and then separated and quantitatively determined by capillary electrophoresis. The buffer electrolyte was prepared on the basis of benzoic acid, diethanolamine, cetyltrimethylammonium bromide and Trilon B. Separation was carried out at a voltage of −20 kV and UV detection at 254 nm.

### 2.3. Genome Analysis

The genome sequence of *L. helveticus* H9 and *L. paracasei* ABK were obtained from the NCBI database (accession numbers JAHLXO010000000.1 and JAHLXM000000000.1, respectively). For the reconstruction of metabolic pathways EC numbers were extracted from genome annotations and were automatically mapped onto the Kyoto Encyclopedia of Genes and Genomes (KEGG) metabolic pathways using KEGG Mapper [25].

### 2.4. Statistical Data Analysis

All statistical comparisons were firstly performed using one-way ANOVA omnibus *F*-Test. When a significant ($p < 0.05$) value of *F*-statistics was found, differences between means were evaluated using Tukey's HSD (honestly significant difference) multiple comparison test ($p < 0.05$).

## 3. Results and Discussion

### 3.1. Composition of Total Amino Acids

The composition of total amino acids in RSM, MRS and postbiotic substances derived from them is presented in Table 1. In general, RSM_Lh and RSM_Lp showed a sharp decrease (about an order of magnitude) in total amino acids compared to RSM, with the concentration of ten amino acids (lysine, valine, phenylalanine, mentioning, histidine, glutamate, proline, serine, tyrosine and alanine) being higher in RSM_Lh than in RSM_Lp. The concentrations of nine amino acids (leucin, isoleucine, threonine, tryptophan, glutamate, glutamine, aspartate, asparagine, arginine) were the same in both RSM_Lh and RSM_Lp, and only the concentration of glycine was higher in RSM_Lp compared to RSM_Lh. It is important to note that although the total amino acid content decreased, the ratio of essential to non-essential amino acids did not change during the fermentation process.

Remarkably, almost no statistically significant changes were observed in MRS_Lh and MRS_Lp, comparing to MRS. The exception was only methionine, which concentration was the same in MRS and MRS_Lh, but increased sharply by almost two and a half times in MRS_Lp. Consequently, most probably during the fermentation of MRS both *L. helveticus* H9 and *L. paracasei* ABK utilized only free amino acids as a nitrogen source.

### 3.2. Composition of Free Amino Acids

The composition of free amino acids for RSM, MRS and postbiotic substances derived from them is presented in Table 2. Since there was more variation in the free amino acid composition compared to the total amino acid composition, the former was additionally represented by the clustered heatmap in Figure 1. Moreover, the constructed heatmap was supplemented with information of the physicochemical properties and common metabolic precursors for each amino acid.

**Table 1.** Composition of total amino acids in RSM, MRS and postbiotic products obtained by their fermentation with *L. helveticus* H9 and *L. paracasei* ABK.

| Amino Acid | Total Amino Acids, mg·(100 mL)$^{-1}$ | | | | | | | | | | | |
| --- | --- | --- | --- | --- | --- | --- | --- | --- | --- | --- | --- | --- |
| | RSM | | | | | | MRS | | | | | |
| | Control | | *L. helveticus* H9 | | *L. paracasei* ABK | | Control | | *L. helveticus* H9 | | *L. paracasei* ABK | |
| | Essential | | | | | | | | | | | |
| | Mean | SD | Mean | SD | Mean | SD | Mean | SD | Mean | SD | Mean | SD |
| Leu + Ile | 296 [a] | 44 | 20 [b] | 3 | 14 [b] | 2 | 177 [a] | 14 | 183 [a] | 15 | 184 [a] | 15 |
| Lys | 138 [a] | 20 | 8.0 [b] | 1.2 | 5.3 [c] | 0.8 | 97 [a] | 15 | 101 [a] | 15 | 109 [a] | 16 |
| Val | 123 [a] | 18 | 9.3 [b] | 1.4 | 6.1 [c] | 0.9 | 102 [a] | 15 | 100 [a] | 15 | 85 [a] | 15 |
| Phe | 106 [a] | 16 | 4.6 [b] | 0.7 | 2.8 [c] | 0.4 | 70 [a] | 3.1 | 70 [a] | 3 | 70 [a] | 3 |
| Thr | 92 [a] | 14 | 6.4 [b] | 0.9 | 5.2 [b] | 0.8 | 49 [a] | 7.0 | 47 [a] | 7 | 46 [a] | 7 |
| Met | 81 [a] | 12 | 8.6 [b] | 1.3 | 3.0 [c] | 0.4 | 39 [a] | 6.0 | 46 [a] | 7 | 111 [b] | 17 |
| His | 83 [a] | 7 | 12 [b] | 1 | 10 [c] | 1 | 56 [a] | 8.0 | 56 [a] | 8 | 55 [a] | 8 |
| Trp | 35 [a] | 2.3 | 0.8 [b] | 0.2 | 0.7 [b] | 0.2 | 3.7 [a] | 0.6 | 4.8 [a] | 0.7 | 4.4 [a] | 0.7 |
| Total: | 953 [a] | 191 | 70 [b] | 14 | 47 [b] | 9 | 594 [a] | 119 | 608 [a] | 122 | 664 [a] | 13 |
| | Non-Essential | | | | | | | | | | | |
| | Mean | SD | Mean | SD | Mean | SD | Mean | SD | Mean | SD | Mean | SD |
| Glu + Gln | 591 [a] | 88 | 28 [b] | 4 | 31 [b] | 5 | 179 [a] | 27 | 153 [a] | 23 | 142 [a] | 21 |
| Pro | 275 [a] | 55 | 43 [b] | 9 | 27 [c] | 6 | 210 [a] | 42 | 189 [a] | 38 | 186 [a] | 37 |
| Asp + Asn | 240 [a] | 36 | 10 [b] | 2 | 14 [b] | 2 | 117 [a] | 18 | 118 [a] | 18 | 97 [a] | 15 |
| Ser | 111 [a] | 7 | 13 [b] | 1 | 4.7 [c] | 0.3 | 65 [a] | 4 | 57 [a] | 4 | 60 [a] | 4 |
| Tyr | 110 [a] | 4.9 | 4.6 [b] | 0.2 | 2.1 [c] | 0.1 | 27 [a] | 4 | 28 [a] | 4 | 7.8 [a] | 1 |
| Ala | 76 [a] | 11 | 19 [b] | 2.9 | 8.9 [c] | 1.3 | 149 [a] | 22 | 163 [a] | 24 | 158 [a] | 24 |
| Arg | 50 [a] | 8 | 2.7 [b] | 0.4 | 3.4 [b] | 0.5 | 63 [a] | 9 | 71 [a] | 11 | 54 [a] | 8 |
| Gly | 35 [a] | 2 | 3.1 [b] | 0.1 | 4.0 [c] | 0.2 | 206 [a] | 9 | 225 [a] | 10 | 228 [a] | 10 |
| Cys | 23 [a] | 3 | ND | - | ND | - | ND | - | ND | - | ND | - |
| Total: | 919 [a] | 184 | 96 [b] | 19 | 64 [c] | 13 | 837 [a] | 167 | 852 [a] | 170 | 791 [a] | 158 |

Means within the same row with different superscripts are significantly different ($p < 0.05$). SD: standard deviation; ND: not detected.

**Table 2.** Composition of free amino acids in RSM, MRS and postbiotic products obtained by their fermentation with *L. helveticus* H9 and *L. paracasei* ABK.

| Amino Acid | Free Amino Acids, mg·(100 mL)$^{-1}$ | | | | | | | | | | | |
| --- | --- | --- | --- | --- | --- | --- | --- | --- | --- | --- | --- | --- |
| | RSM | | | | | | MRS | | | | | |
| | Control | | *L. helveticus* H9 | | *L. paracasei* ABK | | Control | | *L. helveticus* H9 | | *L. paracasei* ABK | |
| | Essential | | | | | | | | | | | |
| | Mean | SD | Mean | SD | Mean | SD | Mean | SD | Mean | SD | Mean | SD |
| Met | ND | - | 4.6 [a] | 1.1 | 2.3 [b] | 0.5 | 27 [a] | 6 | 21 [a] | 5 | 46 [b] | 11 |
| Lys | ND | - | 1.7 [a] | 0.3 | 2.0 [a] | 0.4 | 8.0 [a] | 1.4 | 2.0 [b] | 0.4 | 1.4 [c] | 0.2 |
| Leu + Ile | ND | - | ND | - | 0.56 [a] | 0.1 | 4.6 [a] | 0.8 | 0.29 [b] | 0.05 | 0.46 [c] | 0.08 |
| Trp | ND | - | 0.05 [a] | 0.01 | 0.17 [b] | 0.03 | 3.4 [a] | 0.7 | 0.12 [b] | 0.02 | 0.18 [c] | 0.03 |
| His | ND | - | ND | - | ND | - | 2.4 [a] | 0.5 | ND | - | ND | - |
| Val | ND | - | ND | - | ND | - | 1.6 [a] | 0.3 | ND | - | ND | - |
| Thr | ND | - | ND | - | ND | - | 1.3 [a] | 0.2 | ND | - | ND | - |
| Phe | ND | - | ND | - | ND | - | ND | - | ND | - | ND | - |
| Total: | ND | - | 6.4 [a] | 1.3 | 5.0 [a] | 1.2 | 48 [a] | 10 | 23 [b] | 5 | 48 [a] | 10 |

**Table 2.** *Cont.*

| Amino Acid | Free Amino Acids, mg·(100 mL)$^{-1}$ | | | | | | | | | | | |
| --- | --- | --- | --- | --- | --- | --- | --- | --- | --- | --- | --- | --- |
| | RSM | | | | | | MRS | | | | | |
| | Control | | *L. helveticus* H9 | | *L. paracasei* ABK | | Control | | *L. helveticus* H9 | | *L. paracasei* ABK | |
| | Non-Essential | | | | | | | | | | | |
| | Mean | SD | Mean | SD | Mean | SD | Mean | SD | Mean | SD | Mean | SD |
| Glu + Gln | ND | - | 8.3 [a] | 1.7 | 5.6 [b] | 1.1 | 23 [a] | 5 | 3.3 [b] | 0.6 | 3.5 [b] | 0.7 |
| Asp + Asn | ND | - | 0.50 [a] | 0.10 | 1.1 [b] | 0.2 | 17 [a] | 3 | 0.84 [b] | 0.17 | 0.64 [b] | 0.13 |
| Pro | ND | - | 0.81 [a] | 0.15 | 0.58 [b] | 0.10 | 12 [a] | 2 | 0.69 [b] | 0.12 | 0.90 [b] | 0.16 |
| Arg | ND | - | 1.3 [a] | 0.3 | 1.6 [a] | 0.4 | 5.3 [a] | 1.2 | 1.3 [b] | 0.3 | 1.2 [b] | 0.3 |
| Gly | ND | - | 1.4 [a] | 0.2 | 0.84 [b] | 0.15 | 3.6 [a] | 0.6 | 0.86 [b] | 0.15 | 1.2 [b] | 0.2 |
| Ser | ND | - | 0.51 [a] | 0.09 | 0.11 [b] | 0.02 | 2.8 [a] | 0.5 | 0.24 [b] | 0.04 | 0.22 [b] | 0.04 |
| Ala | ND | - | 0.51 [a] | 0.09 | 0.28 [b] | 0.05 | 2.0 [a] | 0.4 | 0.62 [b] | 0.11 | 0.68 [b] | 0.12 |
| Cys | ND | - | ND | - | ND | - | 1.9 [a] | 0.5 | ND | - | ND | - |
| Tyr | ND | - | ND | - | ND | - | ND | - | ND | - | ND | - |
| Total: | ND | - | 13 [a] | 3 | 10 [a] | 2 | 68 [a] | 14 | 7.8 [b] | 0.16 | 8.3 [c] | 0.2 |

Means within the same row with different superscripts are significantly different ($p < 0.05$). SD: standard deviation; ND: not detected.

Although it was previously reported that fresh milk typically contained on average from 0.06 to 1 mg·(100 mL)$^{-1}$ of each amino acid [26], in our case there were no free amino acids detected in the RSM (Table 3 and Figure 1). Generally, this can be explained by the thermal treatment that that this milk has undergone during the lyophilization procedure [27]. For the postbiotic substances obtained after the RSM fermentation, six amino acids (threonine, valine, histidine, phenylalanine, tyrosine and cysteine) were not detected in either RSM_Lh or RSM_Lp; seven amino acids (methionine, glutamate, glutamine, proline, glycine, seryne and alanine) were approximately twice more abundant in the RSM_Lh; three amino acids (tryptophan, aspartate and asparagine) were approximately twice more abundant in the RSM_Lp; the content of lysine and arginine was the same for both postbiotic substances; and leucin and isoleucine were detected only in the RSM_Lp.

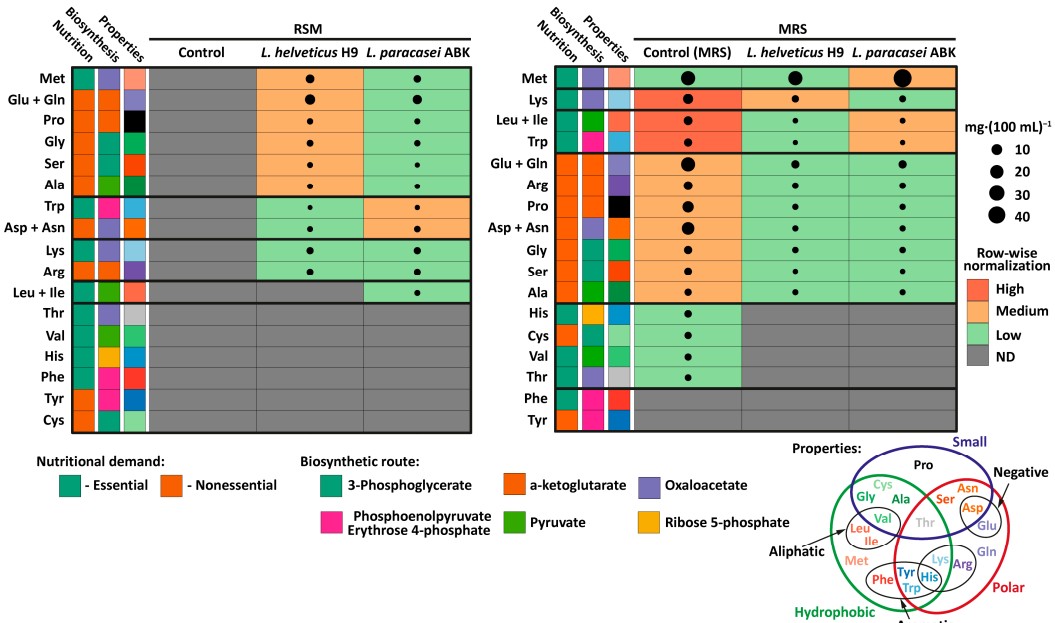

**Figure 1.** Clusters heatmaps depicting the composition of free amino acids in RSM, MRS and postbiotic products obtained by their fermentation with *L. helveticus* H9 and *L. paracasei* ABK. For more detail regarding the color code for physicochemical properties of each amino acid, refer to Heinrich et al. [28].

**Table 3.** Proteolytic systems genes in *L. helveticus* H9 and *L. paracasei* ABK genomes.

| Enzyme | Gene | *L. helveticus* H9 | | *L. paracasei* ABK | |
|---|---|---|---|---|---|
| | | Number of Genes | Locus Number in the Genome | Number of Genes | Locus Number in the Genome |
| CEP proteinases | *prtB* | 1 | MBU6033914 | 1 | MBU6046327 |
| | *prtP* | 1 | MBU6034695 | 1 | MBU6048028 |
| Endopeptidase | *pepO* | 2 | MBU6033720 MBU6034694 | 2 | MBU6046960 MBU6047360 |
| | *pepF* | 1 | MBU6034722 | 3 | MBU6046494 MBU6047890 MBU6047627 |
| | *pepE* | 1 | MBU6034026 | 1 | MBU6047326 |
| Aminopeptidases | *pepC* | 3 | MBU6034983 MBU6034023 MBU6034659 | 1 | MBU6047325 |
| | *pepN* | 1 | MBU6034546 | 1 | MBU6048029 |
| | prolinase, *pepP* | 1 | MBU6035018 | 1 | MBU6046149 |
| | glutaminopeptidase, *pepA* | 1 | MBU6034400 | 0 | no * |
| | proline iminopeptidase, *pepI* | 1 | MBU6034385 | 1 | MBU6046842 |
| | prolidase, *pepQ* | 1 | MBU6034747 | 1 | MBU6046550 |
| | *pepS* | 0 | no | 1 | MBU6047768 |
| Oligo-/Tri-/Di-peptidases | prolinase, *pepR* | 1 | MBU6034760 | 1 | MBU6047007 |
| | tripeptidase, *pepT* | 2 | MBU6034325 MBU6034429 | 0 | no |
| | X-prolil dipeptidyl aminopeptidase, *pepX* | 1 | MBU6034686 | 1 | MBU6046155 |
| | dipepidase, *pepV* | 1 | MBU6034208 | 1 | MBU6047856 |
| | dipepidase, *pepD* | 4 | MBU6034175 MBU6033703 MBU6034538 MBU6034470 | 3 | MBU6046618 MBU6047722 MBU6047002 |
| Total: | | 23 | | 20 | |

no *—not found.

For the postbiotic substances obtained after MRS fermentation, the concentration of all amino acids decreased in both MRS_Lh or MRS_Lp compared to the unfermented medium (Table 3 and Figure 1). The exceptions were phenylalanine and tyrosine, which were absent in MRS, MRS_Lh or MRS_Lp, and methionine, for which the concentration in MRS_Lh was the same as in MRS and increased two-fold in MRS_Lp. Among the amino acids, for which the concentration decreased after fermentation, four amino acids (histidine, cysteine, valine and threonine) totally disappeared both in MRS_Lh and MRS_Lp; nine amino acids (glutamate, glutamine, arginine, proline, aspartate, asparagine, glycine and serine) were present with the same concentration both in MRS_Lh and MRS_Lp; three amino acids (leucine, isoleucine and tryptophan) were present in MRS_Lp at twice the concentration compared to MRS_Lh; and for lysine, its concentration was slightly higher (by 44%) in MRS_Lh compared to MRS_Lp.

Hence, during the RSM fermentation by both strains of probiotic cultures, free amino acids were enriched, and the content of free amino acids in the RSM_Lh postbiotic were somewhat higher than in RSM_Lp. Since amino acids with common biosynthetic precursors did not change their concentration in a similar way during fermentation, it can be assumed

that not amino acid biosynthesis, but extracellular and intracellular proteolytic processes, as well as peptides and amino acid transport inside and outside the cell, are the main causes of the observed changes. Moreover, the lack of correlation between changes in concentrations of amino acids and their physicochemical properties suggests that this process probably involves not one, but several different proteases and transporters with different affinities for substrates.

The in silico analyses of the lactobacillus genomes revealed that the *L. helveticus* H9 genome contains 23 genes encoding proteases and peptidases, while the *L. paracasei* ABK genome contains 20 genes (Table 3). The *L. helveticus* H9 genome contains three genes each encoding aminopeptidase PepC and tripeptidase PepT, four genes encoding dipeptidase PepD, and a gene encoding glutamyl aminopeptidase PepA. At the same time in *L. paracasei*, ABK genome PepT and PepA were not found; however, the presence of three genes encoding PepF oligoendopeptidases, which hydrolyze oligopeptides with a length of seven to seventeen residues, was established. Additionally, the genome of *L. helveticus* H9 and *L. paracasei* ABK contains 73 and 38 putative ABC transporters, which are involved in the transport of oligopeptides and various amino acids (Table S1, Figures S1 and S2). Thus, the genomes of both *L. helveticus* H9 and *L. paracasei* ABK have an effective proteolytic system that comprises cell wall proteinases (CEP), intracellular peptidases and different transporters of amino acids and peptides.

*3.3. Content of Organic Acids*

The content of organic acids is presented in Table 4. With respect to the postbiotic substances obtained from RSM, the content of lactic acid increased by approximately 40-fold after fermentation, and was the same in both RSM_Lh and RSM_Lp. Additionally, the content of acetic acid increased only three times, and succinic acid was detected only in RSM_Lh. With respect to the postbiotic substances obtained from MRS, the lactic acid content in MRS_Lp was slightly higher in MRS_Lh. The concentration of acetic acid in MRS_Lh and MRS_Lp was unaltered compared to MRS, and succinic acid was detected only in MRS_Lh. The observed differences in the composition and content of organic acids in postbiotics can be associated both with differences in the composition of nutrient media and due to the peculiarity of the carbohydrate metabolism of lactobacilli strains.

**Table 4.** Content of organic acids in RSM, MRS and postbiotic products obtained by their fermentation with *L. helveticus* H9 and *L. paracasei* ABK.

| Sample | Content of Organic Acids, mg $(100 \text{ mL})^{-1}$ | | | | | |
| --- | --- | --- | --- | --- | --- | --- |
| | Lactate | | Acetate | | Succinate | |
| **RSM** | | | | | | |
| | Mean | SD | Mean | SD | Mean | SD |
| Control | 36 [a] | 4 | 10 [a] | 1 | ND | - |
| *L. helveticus* H9 | 1371 [b] | 42 | 34 [b] | 3 | 45 | 2 |
| *L. paracasei* ABK | 1133 [b] | 32 | 31 [b] | 2 | ND | |
| **MRS** | | | | | | |
| | Mean | SD | Mean | SD | Mean | SD |
| Control | ND | - | 202 [a] | 11 | ND | - |
| *L. helveticus* H9 | 609 [a] | 22 | 226 [a] | 13 | 13 | 1 |
| *L. paracasei* ABK | 798 [b] | 34 | 217 [a] | 14 | ND | - |

Means within the same column with different superscripts are significantly different ($p < 0.05$). SD: standard deviation; ND: not detected.

*Lactobacillus helveticus* and *Lacticaseibacillus paracasei* are classified as an obligatory homofermentative and as facultative heterofermentative members of *Lactobacillaceae,* respectively [29]. In MRS broth, the sole carbohydrate source is glucose, which can be fermented by Lactobacillus strains to lactic acid by the Embden–Meyerhof–Parnas (glycolysis) path-

way (in homofermentation case) or via phosphoketolase pathway (in heterofermentation case). In glycolysis, the fermentation of 1 mol of glucose leads to the formation of 2 mol of lactate, in the phosphoketolase pathway—only 1 mol of lactic acid and equimolar amounts of carbon dioxide, and acetate or ethanol [30]. In accordance with this and our results (Table 4), we can conclude that both *L. helveticus* H9 and *L. paracasei* ABK strains behave homofermentatively during the MRS broth fermentation.

In the case of RSM medium we have found that both strains of *L. helveticus* H9 and *L. paracasei* ABK produced lactic and acetic acids. Lactate was the main fermentation product of lactose, which is the sole source of carbohydrates in milk. Bioinformatic analyses showed that the lactose is metabolized by *L. paracasei* ABK via tagatose-6P pathway in combination with the Leloir pathway and by *L. helveticus* H9 via a lone Leloir pathway (Table 5). The carbohydrates are preferentially transported by phosphoenolpyruvate-dependent sugar phosphotransferase (PTS) systems. In silico analyses of the *L. paracasei* ABK and *L. helveticus* H9 genomes revealed 50 and 15 PTS systems, respectively, belonging to the Glc, Lac, Man, Fru families (Table S2, Figures S3 and S4), as well as putative ABC transporters in ABK and H9 genomes, respectively, are also involved in the transport of various carbohydrates (Figures S1 and S2). It is believed that the penetration of lactose into the cell thanks to PTS systems, which leads to the phosphorylation of sugars, is associated with the tagatose-6P pathway, while through the permease systems it is associated with the Leloir pathway [29]. Really, in the tagatose-6P pathway a PTS transporter, encoded by the *lacFE* genes and transported lactose, was found in the *L. paracasei* ABK genome and was absent in the genome of *L. helveticus* H9 (Table 4).

**Table 5.** Lactose/galactose metabolism genes in *L. helveticus* H9 and *L. paracasei* ABK genomes.

| Gene | Enzyme Encoded by the Gene | EC: | Locus Number in the Genome | |
| --- | --- | --- | --- | --- |
| | | | *L. helveticus* H9 | *L. paracasei* ABK |
| Tagatose-6P pathway | | | | |
| *lacA* | Galactose-6-phosphate isomerase | 5.3.1.26 | no * | MBU6047228 |
| *lacB* | Galactose-6-phosphate isomerase | 5.3.1.26 | no | MBU6047229 |
| *lacC* | Tagatose 6-phosphate kinase | 2.7.1.144 | no | MBU6047231 |
| *lacD* | Tagatose 1,6-diphosphate aldolase | 4.1.2.40 | no | MBU6047230 MBU6047617 |
| *lacE* | Lactose PTS system EIICB component | 2.7.1.207 | no | MBU6048344 |
| *lacF* | Lactose PTS system EIIA component | 2.7.1.207 | no | MBU6048346 |
| *lacG* | 6-phospho-beta-galactosidase | 3.2.1.85 | no | MBU6048345 |
| *gatY* | Tagatose 1,6-diphosphate aldolase | 4.1.2.40 | no | MBU6048319 |
| Leloir pathway | | | | |
| *galP* *lacS* *lacY* | Galactose permease Galactose–Lactose antiporter Lactose permease | | no | no |
| *lacL* | Beta-galactosidase (GH2) | 3.2.1.23 | MBU6035121 | MBU6048422 |
| *lacM* | Beta-galactosidase | 3.2.1.23 | MBU6035122 | MBU6048423 |
| *galM* | Galactose epimerase | 5.1.3.3 | MBU6034706 | MBU6047737 |
| *galK* | Glucokinase | 2.7.1.6 | MBU6034704 | MBU6047741 |
| *galT* | Galactose-1-phosphate uridylyltransferase | 2.7.7.12 | MBU6034705 | MBU6047739 |
| *galE* | UDP-glucose-4-epimerase | 5.1.3.2 | MBU6035123 | MBU6047740 |
| *pgm* | Phosphoglucomutase | 5.4.2.2 | MBU6034768 | MBU6046814 |

no *—not found.

Entered into the cell, the lactose or lactose-6P, depending on the transport system, are hydrolyzed by β-galactosidase (EC: 3.2.1.23) or 6-phospho-β-galactosidase (EC: 3.2.1.85), which are encoded by the *lacZ/lacLM* and *lacG* genes, respectively, in lactobacilli genomes. The *lacG* gene is found only in the *L. paracasei* ABK genome and is absent in *L. helveticus* H9 (Table 4). Indeed, *lacG* genes have been found in all genomes with tagatose-6P pathway [29]. On the contrary, *lacLM* genes are found in both *L. paracasei* ABK and *L. helveticus* H9 genomes (Table 4). Interestingly, we did not find classical lactose/galactose permeases (non-PTS permease) such as *galP*, *lacS* and *lacY* in the ABK and H9 genomes. However, these putative genes are not always present in all genomes possessing the Leloir pathway [31]. It is possible that the transport of lactose into the bacterial cell for metabolism via the Leloir pathway may be carried out by other hypothetical transport systems. The end products of the tagatose-6P pathway and the Leloir pathway are glyceraldehyde-3-phosphate (GPDH) and glucose-1-phosphate (Glucose-1P), respectively. These products then enter the glycolysis pathway to form pyruvate, which is then converted into L- and D-lactate by the lactate dehydrogenases (EC: 1.1.1.27 and 1.1.1.28, respectively). The enzymatic pathway of *L. helveticus* H9 and *L. paracasei* ABK for pyruvate metabolism is presented in Figure 2.

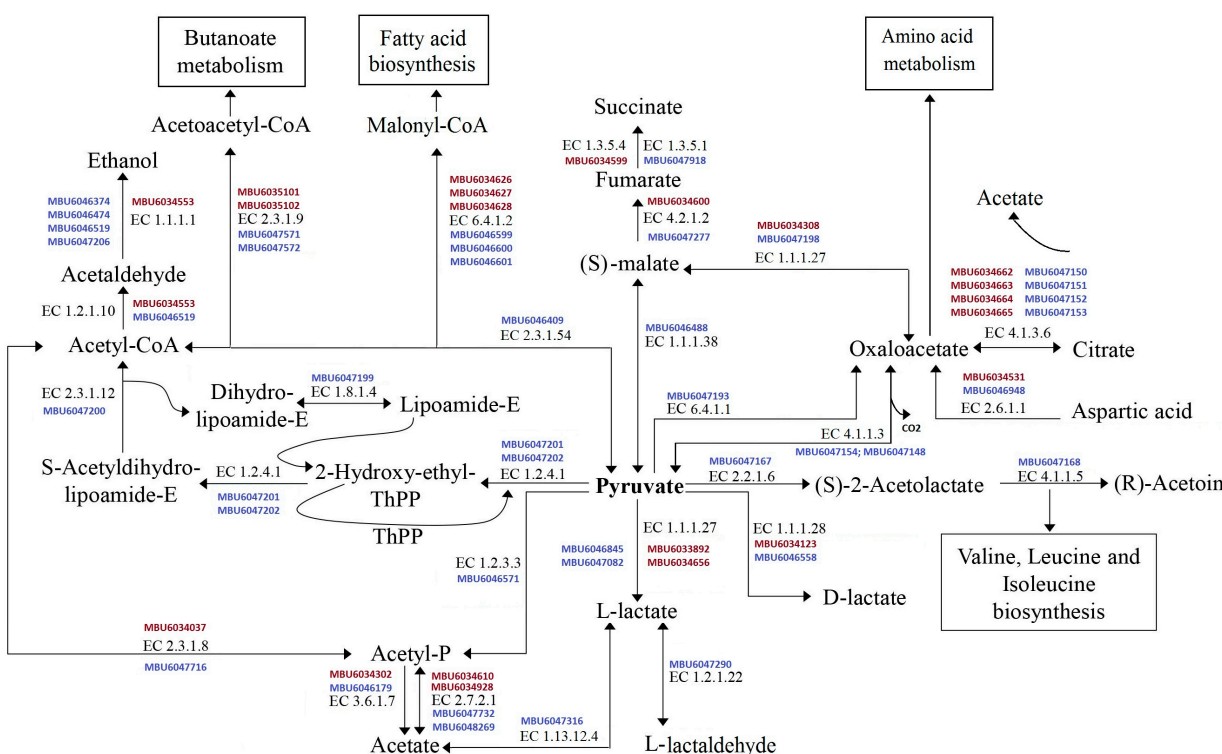

**Figure 2.** Reconstruction of the pyruvate metabolic pathway of *L. helveticus* H9 (colored in red) and *L. paracasei* ABK (colored in blue), according to the De Angelis et al. [32], with modifications.

The presence of acetate in the RSM_Lh and RSM_Lp postbiotic substances can be the result of the different biochemical pathway, e.g., the degradation product of the produced lactic acid, and/or the result of the citrate metabolism [5,6], while in the MRS broth the H9 and ABK strains showed an unambiguous homofermentative acid profile (Table 3). In the SRM medium the character of fermentation of these strains changed from a homolactic to mixed-acid profile, which is explained by the presence of citrate in the milk. Similar results are shown in the work of Zalán et al. for some *Lactobacillus* strains [33]. In MRS_Lh and RSM_Lh we also found succinic acid in low concentration (Table 3), which can be explained by the citrate utilization of *L. helveticus* H9 in the RSM. In the absence of oxaloacetate decarboxylase (OAD, EC: 4.1.1.3) in genome (Figure 2), citrate utilization by *L. helveticus* H9 follows through the succinic acid pathway, which is quite common among lactobacilli [33]. At the same time, the OAD genes were found in the *L. paracasei* ABK genome (*oadA* and

*oadB*, Figure 2), so only acetic acid is detected (Table 4). However, on the MRS medium, due to inhibition of citrate utilization by glucose, the succinate can also be produced by *L. helveticus* H9 from aspartic acid (Figure 2).

## 4. Conclusions

In this work, the amino and organic acid composition of complex postbiotic substances obtained on the basis of probiotic cultures of *L. helveticus* H9 and *L. paracasei* ABK using the nutrient medium MRS-broth and reconstructed skimmed milk (RSM) was evaluated. It was found that in the RSM-based postbiotic substances, the ratio of essential to non-essential amino acids in the total amino acid composition was unchanged, while the medium was enriched in free amino acids. For the MRS-based postbiotic substances obtained with *L. paracasei* ABK, the two-fold increase in both total and free methionine content was observed, while the changes in the concentrations of other amino acids were negligible. Furthermore, organic acids (lactic, acetic, succinic) were found in the studied probiotic substances. The conducted studies have shown that organic acids are a component of metabolic processes occurring in postbiotic substances obtained using the studied probiotic cultures of *L. paracasei* ABK and *L. helveticus* H9. However, the composition and concentration of organic acids depends on the medium used. So, in the studied samples, succinic acid was found only in samples obtained using the probiotic culture *L. helveticus* H9, while in the samples obtained using the probiotic culture *L. paracasei* ABK, succinic acid was not detected. At the same time, the content of acetic acid was maximum in the samples obtained using the studied probiotic cultures of *L. paracasei* ABK and *L. helveticus* H9, where the nutrient medium MRS-broth (MRS) was used as a medium for the accumulation of metabolites. The results of the work suggested that postbiotic substances obtained on the basis of *L. helveticus* H9 and *L. paracasei* ABK can later be used to obtain complex postbiotic food supplements and be used as part of dietary supplements and to enrich food products and animal feed with amino acids, including essential amino acids.

**Supplementary Materials:** The following supporting information can be downloaded at: https://www.mdpi.com/article/10.3390/fermentation9050460/s1, Figure S1: ABC transport systems of *L. paracasei* ABK; Figure S2: ABC transport systems of *L. helveticus* H9; Figure S3: Phosphotransferase systems (PTSs) of *L. paracasei* ABK; Figure S4: Phosphotransferase systems (PTSs) of *L. helveticus* H9; Table S1: Oligopeptide, di-/tripeptide and amino acid transport systems of *L. helveticus* H9 and *L. paracasei* ABK; Table S2: Carbohydrate transport systems of *L. helveticus* H9 and *L. paracasei* ABK.

**Author Contributions:** Conceptualization, I.V.R. and E.A.Y.; methodology, I.V.R. and E.A.Y.; validation, I.V.R.; formal analysis, I.V.R. and E.A.Y.; investigation, E.A.Y. and V.A.L.; data curation, I.V.R. and E.A.Y.; writing—original draft preparation, I.V.R., E.A.Y. and V.A.L.; writing—review and editing, I.V.R. and E.A.Y.; visualization, E.A.Y. and V.A.L.; supervision, I.V.R.; funding acquisition, I.V.R. All authors have read and agreed to the published version of the manuscript.

**Funding:** This research received no external funding.

**Institutional Review Board Statement:** Not applicable.

**Informed Consent Statement:** Not applicable.

**Data Availability Statement:** The data presented in this study are available in the article and Supplementary Material.

**Acknowledgments:** We express our thanks to Konstantin V. Moiseenko and Tatiana V. Fedorova from the A.N. Bach Institute of Biochemistry, Research Center of Biotechnology, Russian Academy of Sciences, who provided invaluable assistance in the bioinformatic analysis of lactobacillus genomes.

**Conflicts of Interest:** The authors declare no conflict of interest. The funders had no role in the design of the study; in the collection, analyses, or interpretation of data; in the writing of the manuscript, or in the decision to publish the results.

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
