# Peer review of "Evaluation of the Amino Acid Composition and Content of Organic Acids of Complex Postbiotic Substances Obtained on the Basis of Metabolites of Probiotic Bacteria Lacticaseibacillus paracasei ABK and Lactobacillus helveticus H9"

_fermentation, doi:10.3390/fermentation9050460_

Round 1

Reviewer 1 Report

Corrections are attached.

Author Response

Dear reviewer, our team would like to thank you for your comments and time employed in reviewing this manuscript. The manuscript was considerably rewritten, according to your suggestions.

Reviewer 2 Report

 The objective of the present research was evaluated the composition of an enriched skimmed milk by the fermentation by two different lactobacilli: Lactobacillus paracasei ABK and Lactobacillus helveticus H-9. Specially, the presence of free amino acids, essential and no essential, and the acid composition was determined. The MRS-broth media was used as a control, nevertheless, it was a very different composition than the skimmed milk. In the same media each lactobacill enriched the postbiotic with different amino acid and organic acid.

Why used the MRS media as a control when it will not be used as a food matrix for commercialized postbiotic products?

In the manuscript, some mistakes should be corrected and other questions need more details:

Abstract:

The antimicrobial activity of the postbiotic substances was no evaluated. It should be deleted from the L.10.

L.20 add its and delete -s, because you are talking about MRS media. In its postbiotic.

L27. Delete ‘.’ after broth and all the parenthesis in the sentence. Also eliminate the complete name of the bacteria.

L30 and 31. Delete the parenthesis from the numbers

L33. Add the final point

Introduction:

L55. Specify which environmental conditions are you mean (stomach acid..enzymes..)

L71. Unific short-chain or short chain in all the text (L73, L75)

L99. Correct: Non-essential amino acids

L120. All microorganisms should write in cursive

L123. Add the reference of Toe et al.

Objects and methods

The way how you obtained and maintain the microorganism should be described in this section. Also, the initial inoculation level and the final population after 24 hours and the incubation conditions should be detailed: static growth, in air conditions or in a modified atmosphere...

L137. The skimmed milk is form cow? Detail it

L141. Added microorganism growth conditions before inoculate the MRS-broth and skimmed milk, inoculation levels, etc

Add a statistical analyse section

Results

A statistical analyse is necessary to determine if the increase or decrease of amino acid are significant in table 1 and 2.

L184. Eliminate hyphen before MRS

Table 1. Express units as mg/100 ml

L200. Delete in between: , in mg/100

Table 2. In the table header correct (MRS) to (SM)

Figure 1. Translate the vertical axis title

The information shown in figure 1 it’s a duplication from the table 1 information.

L203. When you take into account the results deviation, non-difference was observed. A statistical analyse should be included.

L218. Non noticed the higher increase of methionine by L. paracasei!!

L222. Similar results were observed by other authors? Please discuss the obtained results.

L232 and 234. Add cursive in de novo

L233. Add cursive in silico

L237. In figure 1 an increase of tryptophan and lysine was observed, how it is possible if one enzyme is missing?

L264. Delete ABK

Data shown in table 5 and table 7 could appear together and detail the number of samples used to do the average. Thus, the new average will appear in the text line 295 and so on. Idem in the table 4 and 6.

Table 4, 5, 6 and 7: Correct the table header units: mg/100 ml. Also, a statistical analyse should be done in the organic acid content to compare control and the obtained postbiotics.

Table 5 correct the sample name: Com to SM. Also, it in line 294.

After observe the huge difference between strains in organic acid content production, why not was done a bioinformatic analyse as it was done in the amino acid biosynthesis?

 Please compare the obtained results with the previous observed by other authors.

Conclusion

L332. No significant differences were observed in all the amino acids enumerated.

L333. Review the results, because the higher increase was not in the glycine (tryptophan in H9 and methionine in ABK).

Author Response

Dear reviewer, our team would like to thank you for your comments and time employed in reviewing this manuscript. The manuscript was considerably rewritten, according to your suggestions: all the necessary data were added into the “Materials and Methods” section; statistical data analysis was performed; the increase of methionine was pointed out; the bioinformatic search for the genes involved in the metabolism of organic acids was performed; and the overall readability was increased.

Reviewer 3 Report

Dear authors,

The manuscript is interesting but full of mistakes.

Postbiotics have high bioavailability since they reach the colon by 95-97 % - there are many criteria to declare something has high bioavailability and that it reaches colon in high amount unchanged is only one or them.

- Do postbiotics contain SCFA? OR are they products of digestion of postbiotics by bacteria?

- Are SCHF really the basis of antimicrobial activity of LAB? Primarily it is considered that these are food for colonocytes.

- Line 92: This is such an old reference (16) to be cited in that place!

Line 99: you probably meant non essential?!

- Please put bacteria in italics - Line 120

Line 121: What do you mean with type of bacteria? do you mean genus?

Line 126: It is weird to write it is probiotic property their antibacterial activity

Figure 1: The ordinate axis is in russian. But why didn't you present data in the same way for the skimmed milk?

What is the difference between table 6 and 4?

What is the difference between table 5 and 7?! It is not clear. 

In conclusion you could mention the potential this has. Where could these results be used.. Did you mean food supplements for animal feed or humans?

Author Response

Dear reviewer, we appreciate your attention to our manuscript and are grateful for the valuable recommendations. Thank you for the careful review of the manuscript. Were delete all confusing information from the “Introduction” section of the manuscript. The figure was improved. The overall readability was improved. The “Conclusion” as well as almost all “Results and Discussion” were rewritten.

Round 2

Reviewer 2 Report

Dear authors. I have appreciated the modifications in the manuscript (clearer abstract, a detailed method and included a statistic analyse of the results). Congratulations. I have added the pdf and I detailed some mistakes on it, these should be applied to the manuscript (write in italic the microorganism, and unify the format in the reference section).   

Reviewer 3 Report

The manuscript has been substantially improved.

I recommend its publication in present form.